# PhenoBrain: Phenotype-Conditioned Long-Range Communication for Multi-Modal Brain Network Analysis

Lingyuan Meng [1]  Ke Liang [†1]  Hao Li [1]  Meng Liu [2]  Weijia Shi [1]  Miaomiao Li [3]  Yang Gao [4]  Xinwang Liu [†1]

## Abstract

Multi-modal brain network analysis aims to predict neuropsychiatric status from functional connectomes with heterogeneous phenotypes. However, most existing methods treat phenotypes as auxiliary features and perform late fusion, implicitly assuming that the connectome representation should be learned in the same way regardless of phenotype. In clinical neuroscience the same functional connectivity pattern may support different conclusions under different phenotype contexts. To bridge this gap, we propose **PhenoBrain**, a novel framework for multi-modal brain network analysis that injects phenotype information at the mechanism level rather than only at the classifier level. Specifically, we propose a phenotype-conditioned long-range routing mechanism, which learns a subject-specific multi-hop communication kernel to model long-range connectome interactions. Furthermore, we propose a phenotypic-guided attention mechanism regulation method, which uses phenotypic information as a conditional prior to regulate the learning process of attention in brain networks. To verify our method, we constructed two multi-modal brain network analysis datasets. Extensive experiments demonstrate that PhenoBrain achieves state-of-the-art performance.

## 1. Introduction

Brain network analysis utilizes graph modeling of brain imaging data, where nodes represent regions of interest (ROIs) and edges represent the connection strength between ROIs (Luo et al., 2024; Yang et al., 2025a;b). It plays a crucial role in early screening, patient stratification, and risk assessment of neurological diseases (Cè et al., 2023; Topiwala et al., 2023). However, due to the specificity of an individual's developmental stage, symptom spectrum, and treatment context, the same connectivity abnormality often has different meanings in different individuals (Liu et al., 2024; 2025; Meng et al., 2023). This has prompted research into phenotypic multimodal brain network analysis, which jointly models connectomes with structured phenotypic variables to make accurate and clinically evidence-based predictions (Tong et al., 2023; Yu et al., 2024; Zhuo et al., 2023; **?**). Importantly, this approach differs from the multimodal concept commonly found in neuroimaging, which integrates multiple imaging modalities or multiple connectivity types. Imaging multimodal analysis primarily focuses on similar time scales and struggles to encompass the clinical context itself, while phenotypic information provides contextual complementarity, guiding how to extract evidence from the same functional network.

Although phenotypic multimodal approaches are closer to clinical tasks, existing methods still face two key challenges. **(1) Fusion drawbacks**. Figure 1.(c) show that existing methods fuse phenotypic vectors before and after the classifier, which is equivalent to assuming that brain network representation learning does not need to change with phenotypic changes. However, in multimodal brain network analysis, phenotypic should play a contextual role, determining which connections should be considered evidence and how information should propagate along long-range pathways. In other words, existing methods are more about fusing decision-making conclusions than fusing evidence generation mechanisms. **(2) Ignoring long-range correlations**. Figure 1.(a) show that, in the neuroscience field, brain function is often achieved not through small changes near a certain ROI, but through long-range cooperation such as inter-network coupling and cross-hemispheric synchronization. For example, memory function mainly depends on the cooperation between the cerebral cortex and the hippocampus, which are far apart in the brain. Traditional graph learning methods mainly focus on local neighborhood aggregation, making it difficult to model the cooper-

[1]Colledge of Computer and Science, National University of Defense Technology, Changsha, China [2]College of Artificial Intelligence, Henan University, Zhengzhou, China [3]School of Computer Science and Engineering, Changsha University, Changsha, China [4]School of Intelligence Science and Technology, Nanjing, China. Correspondence to: Ke Liang <liangke200694@126.edu>, Xinwang Liu <xinwangliu@nudt.edu.cn>.

*Proceedings of the 43rd International Conference on Machine Learning*, Seoul, South Korea. PMLR 306, 2026. Copyright 2026 by the author(s).

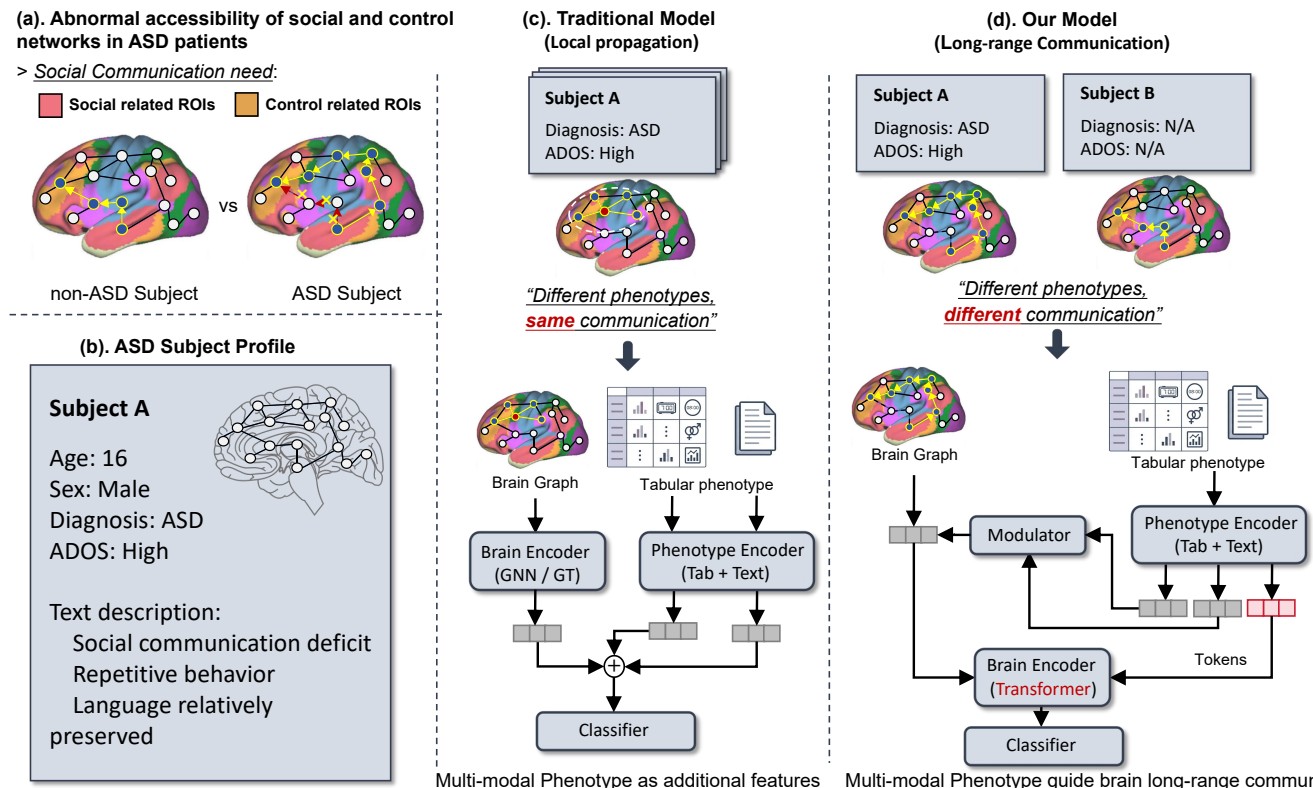

*Figure 1.* An illustration of our motivation. Subgraph (a) illustrates the accessibility of social and control networks in healthy individuals and individuals with autism spectrum disorder (ASD). Compared to healthy individuals, the social and control networks of ASD patients require longer paths to reach. Subgraph (b) shows phenotypic and textual data samples from ASD patients. Subgraphs (c) and (d) compare traditional multimodal brain network analysis models with our proposed method, respectively. Specifically, traditional methods employ a same local aggregation approach for different phenotypes, while our method guides individuals to engage in personalized long-distance communication through phenotype analysis.

ation of most important brain regions. Therefore, phenotypic multimodal brain network analysis urgently needs a mechanism-level framework that allows phenotypes to enter the connectome representation learning process as conditional variables, while explicitly modeling interpretable long-range communication and learning phenotype-specific ROI interaction patterns at the attention level, thus forming a consistent closed loop in terms of performance, generalization, and interpretability. These challenges have prompted us to construct a phenotypic conditional multimodal brain network analysis framework that can model long-range correlations while truly learning personalized, interpretable brain network representations.

Therefore, we propose PhenoBrain, a conditionalized phenotypic multimodal brain network analysis framework. Our core insight is that structured phenotypic data and natural language descriptions are not additional features, but should guide the model in conveying information and selecting evidence within the brain network. Thus, PhenoBrain injects phenotypic information from the representation learning stage to address both the challenges of insufficient long-

range dependency modeling and coarse late-stage fusion. Specifically, we first design a Phenotype-Conditioned Long-Range Router, constructing a multi-hop routing kernel on the brain map and generating individual-specific long-range communication representations with low-rank, phenotypic-related modulation, thereby explicitly capturing long-range associations across the network. Secondly, we propose the Phenotype Prompt Generator, which converts phenotypic descriptions into phenotype prompt tokens and inputs them, along with ROI representations, into the Graph Transformer. Furthermore, we leverage phenotypic context to modulate the Transformer's attention, allowing different individuals to exhibit different attention patterns in ROI-ROI and prompt-ROI interactions, achieving individualized fusion at the mechanistic level. Our main contributions are summarized as follow:

- We propose a novel phenotype-conditioned multi-modal brain network analysis framework, termed PhenoBrain, which guides brain map aggregation through multi-modal phenotype information, enabling personalized brain network analysis.

- We propose a novel multimodal brain network fusion strategy, converting multimodal phenotypic information into prompt tokens to guide the attention learning of the graph-transformer, enabling the model to query and aggregate ROI evidence in a probe-like manner.

- We construct two phenotype-based multimodal brain network analysis datasets based on existing image data, demonstrating the superiority and robustness of PhenoBrain through extensive experiments.

## 1.1. Related Work

### 1.1.1. MULTI-IMAGE MULTI-MODAL BRAIN NETWORK ANALYSIS

Multi-image multi-modal brain network analysis has been a critical area of research in neuroimaging, where different imaging modalities, such as fMRI, EEG, and structural MRI, are integrated to study brain networks. Several methods have explored the fusion of multi-modal imaging data to improve the understanding of brain connectivity and its relationship to various disorders. MMGL (Zheng et al., 2022) is a method that employs multi-view learning to integrate multiple brain graphs obtained from different imaging modalities. MMGL focuses on clustering brain networks by learning a shared latent space across views, aiming to uncover coherent brain regions that correlate with specific functional states or disease conditions. CroGen (Luo et al., 2022) takes a similar approach, but it also generates missing network information across views. By filling in missing data, CroGen can better handle incomplete multi-modal datasets, improving the accuracy of brain network analysis for disorder diagnosis. RTGNN (Zhao et al., 2022) extends graph neural networks by incorporating tensor-based multi-view data. RTGNN learns the joint representation of multi-modal brain networks through reinforced aggregation techniques, enhancing the model's ability to capture long-range dependencies and multi-scale patterns across different modalities. Cross-GNN (Yang et al., 2023) uses a cross-modal mutual learning approach to map multi-modal brain connectomes, where the model learns to align representations from different modalities to improve classification performance for brain disorders. This method shows significant improvement in diagnosis accuracy by learning a joint representation of structural and functional connectivity. While these methods focus on integrating multi-modal imaging data for brain network analysis, they largely assume that the imaging modalities are the primary sources of information. This overlooks the potential of integrating phenotypic data, such as clinical and behavioral characteristics, which are crucial for understanding individualized brain network patterns and diseases.

### 1.1.2. PHENOTYPIC-DRIVEN MULTI-MODAL BRAIN NETWORK ANALYSIS

Phenotypic-driven multi-modal brain network analysis, which incorporates clinical and behavioral information alongside neuroimaging data, is an emerging research area. These approaches aim to better personalize brain network models by accounting for the complex interplay between brain structure, function, and the individual's phenotype. MGDR (Jiang et al., 2024) introduces a disentangled representation learning framework that separates different sources of information, such as imaging and phenotypic data. By learning distinct representations for different modalities, MGDR can better predict brain diseases, improving interpretability and predictive power. This method focuses on making the network predictions more robust by disentangling the contributions of different data sources. SPromptGL (Wan et al., 2025) takes a novel approach by using semantic prompts to guide graph learning. This method leverages textual phenotype descriptions as prompts to help the model focus on relevant brain regions and connectivity patterns. By incorporating semantic prompts into graph learning, SPromptGL improves the model's ability to capture phenotype-related variations in brain network structures, aiding in the diagnosis of brain diseases. Despite the promise of these methods, they still face limitations in fully integrating phenotypic data into the brain network learning process. Current approaches often treat phenotypic data as a separate input during the late fusion stage or use simplistic models that do not fully capture the complex relationships between phenotype and brain connectivity. Our approach overcomes these limitations by introducing a mechanism-level fusion framework, which allows phenotypic information to directly influence how brain network representations are learned, leading to more individualized and interpretable brain network models.

## 2. Methodology

### 2.1. Preliminaries

In our multi-modal brain network analysis method, each subject $s \in \{1, \ldots, S\}$ is associated with (i) an fMRI-derived functional brain network, (ii) structured phenotype variables, and (iii) a free-form phenotype description. Concretely, the functional connectome is represented as a graph $G^{(s)} = (V, \mathbf{A}^{(s)}, \mathbf{x}^{(s)})$, where $V = \{1, \ldots, N\}$ denotes $N$ regions-of-interest, $\mathbf{A}^{(s)} \in \{0, 1\}^{N \times N}$ is the adjacency matrix, and $\mathbf{X}^{(s)} \in \mathbb{R}^{N \times d}$ denotes the node feature matrix. The structured phenotype is a tabular vector $\mathbf{t}^{(s)} \in \mathbb{R}^{d_t}$, and the textual phenotype is a token sequence $p^{(s)} = (w_1, \ldots, w_L)$. Given labels $y^{(s)}$, our goal is to learn a classification predictor $\hat{y}^{(s)} = f(G^{(s)}, t^{(s)}, p^{(s)})$.

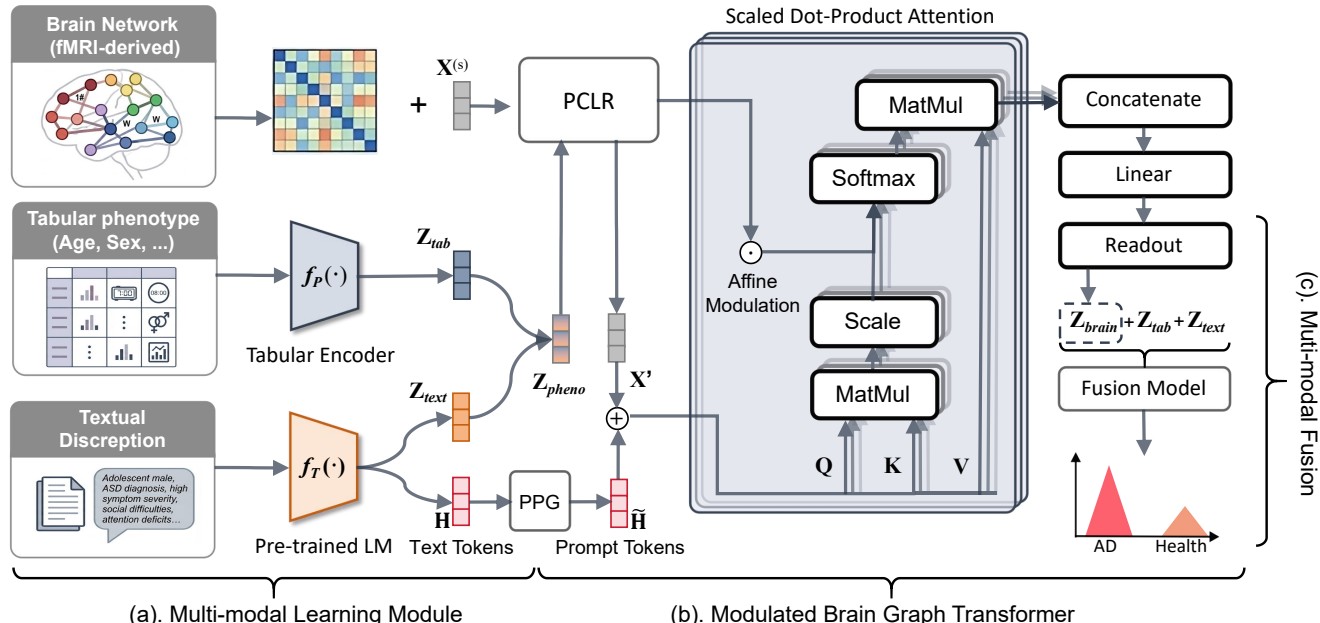

*Figure 2.* The framework of the proposed PhenoBrain. PhenoBrain contains three major components, *i.e.*, (a) multi-modal learning module, (b) a modulated brain graph transformer, and (c) a multi-modal fusion head.

## 2.2. Overview of PhenoBrain

Our PhenoBrain contains three major components, *i.e.*, (a) multi-modal learning module that encodes tabular and textual phenotypes into a global phenotype context vector $\mathbf{z}_{\text{pheno}}^{(s)}$ and a set of phenotype prompt tokens $\tilde{\mathbf{H}}^{(s)}$; (b) a modulated brain graph transformer that performs phenotype-conditioned long-range routing over the connectome to obtain long-range node representations and conducts prompted and modulated self-attention over ROI tokens and phenotype prompt tokens;(c) a multi-modal fusion head that integrates the brain-network embedding with phenotype embeddings for final prediction. Figure 2 illustrates the proposed framework of the proposed PhenoBrain. Note that in our multi-modal framework, phenotypes condition the mechanisms of connectome representation learning, rather than being fused only at the decision layer.

## 2.3. Multi-modal Learning Module

### 2.3.1. MULTI-MODAL DATA PREPARING

**Brain network construction.** For each subject, we first compute the functional connectivity (FC) matrix $\mathbf{C}^{(s)} \in \mathbb{R}^{N \times N}$, *i.e.*, the Pearson correlation between ROI time series. After that, we form a sparse graph by thresholding FC values:

$$\mathbf{A}_{ij}^{(s)} = \mathbb{1}\left(\left|\mathbf{C}_{ij}^{(s)}\right| \geq \tau\right), \quad i \neq j, \tag{1}$$

where $\tau$ is a threshold controlling graph sparsity. Node features $\mathbf{X}^{(s)}$ can be instantiated by ROI-level time-series statistics or by connectivity-derived descriptors. Note that

our method is agnostic to the specific choice, as long as $\mathbf{X}^{(s)} \in \mathbb{R}^{N \times d}$.

**Tabular and textual phenotypes.** In our paper, the structured phenotypic data is native to the dataset and primarily covers long-term characteristics of the instances, including development, symptom spectrum, treatment history, and etc. Structured phenotypes $t^{(s)}$ are normalized per feature and missing values are handled by standard imputation with missingness indicators. The textual phenotype $p^{(s)}$ is provided in the form of clinical narratives and is a natural language description of the tabel phenotype, constructed by domain experts.

Note that, all input PPG texts were preprocessed to remove keywords containing disease diagnosis conclusions (such as 'ASD', 'Diagnosis') or total scale scores (such as 'ADOS Total'), ensuring that the texts contained only phenotypic descriptions related to behavioral observations, language abilities, etc.

### 2.3.2. MULTI-MODAL FEATURE ENCODING

In order to encode multi-modal data, we utilize different encoders. Specifically, we encode tabular phenotypes via an MLP:

$$\mathbf{z}_{\text{tab}}^{(s)} = f_p\left(t^{(s)}\right) \in \mathbb{R}^{d_P}, \tag{2}$$

where $d_p$ is the feature dimention. Furthermore, we use a pre-trained language model $f_t(\cdot)$ to obtain both a global text

embedding and token-level embeddings:

$$\mathbf{z}_{\text{text}}^{(s)}, \mathbf{H}^{(s)} = f_t\left(p^{(s)}\right), \qquad (3)$$

where $\mathbf{z}_{\text{text}}^{(s)} \in \mathbb{R}^{d_\ell}$ is the pooled representation and $\mathbf{H}^{(s)} \in \mathbb{R}^{L \times d_\ell}$ are token embeddings. Finally, we combine tabular and textual phenotypes to form a global conditioning vector:

$$\mathbf{z}_{\text{pheno}}^{(s)} = \phi\left([\mathbf{z}_{\text{tab}}^{(s)} \| \mathbf{z}_{\text{text}}^{(s)}]\right) \in \mathbb{R}^{d_c}, \qquad (4)$$

where $\phi(\cdot)$ is a MLP and $[\cdot \| \cdot]$ denotes concatenation.

## 2.4. Modulated Brain Graph Transformer

The Modulated Brain Graph Transformer aims to extract a phenotypic-aware brain network representation from the functional connectivity graph, allowing the model's focus on key ROIs and long-range pathways to vary with individual phenotypes. Note that this module uses phenotype as a conditional variable to alter the brain network's feature learning mechanism, rather than treating phenotype as the final concatenated feature. Concretely, the modulated connectome encoder consists of three components, *i.e.*, (i) phenotype-conditioned long-range routing (PCLR), (ii) phenotype prompt generation (PPG), and (iii) phenotype-modulated scaled dot-product attention.

### 2.4.1. PHENOTYPE-CONDITIONED LONG-RANGE ROUTING

In functional connectivity networks, clinically relevant signals are often manifested in long-range pathways across networks regions. However, the learning process of conventional graph neural networks or Transformers does not explicitly distinguish between long-range communication and short-range adjacency. The goal of PCLR is to construct an interpretable long-range communication kernel and phenotypically condition it, thereby explicitly modeling how information should travel long distances in the brain network within the phenotypic context of the subject.

Specifiaclly, we first define a base ROI-to-ROI communication strength matrix $\mathbf{F}_{\text{base}}^{(s)} \in \mathbb{R}^{N \times N}$ from FC, which captures subject-specific coupling magnitudes, while $\mathbf{A}^{(s)}$ enforces sparsity. After that, in order to make long-range communication *phenotype-conditioned*, we generate a learnable low-rank adjustment:

$$\Delta \mathbf{F}^{(s)} = \mathbf{U} \operatorname{diag}\left(\mathbf{W}\mathbf{z}_{\text{pheno}}^{(s)}\right) \mathbf{V}^\top, \qquad (5)$$
$$\mathbf{U}, \mathbf{V} \in \mathbb{R}^{N \times r}, \ W \in \mathbb{R}^{r \times d_c}, \ r \ll N.$$

After that, we then obtain the phenotype-conditioned communication map matrix as follow:

$$\mathbf{F}^{(s)} = \operatorname{softplus}\left(\mathbf{F}_{\text{base}}^{(s)} + \Delta \mathbf{F}^{(s)}\right) \in \mathbb{R}^{N \times N}. \qquad (6)$$

Then, we define a column-normalized routing kernel:

$$\mathbf{R}^{(s)} = \left(\mathbf{F}^{(s)} \odot \mathbf{A}^{(s)}\right)\left(\mathbf{D}^{(s)}\right)^{-1} \in \mathbb{R}^{N \times N}, \qquad (7)$$
$$\mathbf{D}_{jj}^{(s)} = \sum_{i=1}^{N} \left(\mathbf{F}^{(s)} \odot \mathbf{A}^{(s)}\right)_{ij}.$$

Here $\odot$ denotes the Hadamard product. The normalization makes each column of $\mathbf{R}^{(s)}$ sum to 1. Then, we build long-range node features via multi-hop propagation:

$$\mathbf{E}_{\text{LR}}^{(s)} = \sum_{k=1}^{K} \alpha_k \left(\mathbf{R}^{(s)}\right)^k \mathbf{X}^{(s)} \in \mathbb{R}^{N \times d}, \qquad (8)$$

where $\alpha_k = \operatorname{softmax}(\eta)_k$, $\alpha_k \geq 0$. Actually, $\sum_{k=1}^{K} \alpha_k = 1$. Finally, we inject long-range information into ROI tokens:

$$\mathbf{X}'^{(s)} = \mathbf{X}^{(s)} + \mathbf{E}_{\text{LR}}^{(s)}\mathbf{W}_{\text{lr}} \in \mathbb{R}^{N \times d}, \quad \mathbf{W}_{\text{lr}} \in \mathbb{R}^{d \times d}. \qquad (9)$$

### 2.4.2. PHENOTYPE PROMPT GENERATION

Textual phenotypes contain rich but unstructured semantic information, which can easily lose conceptual differences if compressed into a single global vector. PPG aims to compress key semantics from text into a set of learnable prompt tokens, serving as the input graph Transformer for semantic probes. Specifically, we convert token embeddings $\mathbf{H}^{(s)} \in \mathbb{R}^{L \times d_\ell}$ into a compact set of $M$ phenotype prompt tokens $\tilde{\mathbf{H}}^{(s)} \in \mathbb{R}^{M \times d}$ using a prompt generator:

$$\tilde{\mathbf{H}}^{(s)} = \operatorname{PPG}\left(\mathbf{H}^{(s)}, \mathbf{z}_{\text{pheno}}^{(s)}\right). \qquad (10)$$

Intuitively, each prompt token acts as a semantic probe that queries the connectome under the phenotype context.

### 2.4.3. PHENOTYPE-MODULATED SCALED DOT-PRODUCT ATTENTION

The goal of Phenotype-modulated Scaled Dot-Product Attention is to allow phenotype to directly influence the Transformer's attention scoring mechanism, enabling the model to learn phenotype-specific interaction patterns. Firstly, we form the input token matrix by concatenating phenotype prompts and ROI tokens:

$$\mathbf{Z}^{(s,0)} = [\tilde{\mathbf{H}}^{(s)} \| \mathbf{X}'^{(s)}] \in \mathbb{R}^{(M+N) \times d}. \qquad (11)$$

Let $L_{\text{tr}}$ denote the number of Transformer layers and $H$ the number of attention heads ($d_h = d/H$). For each layer $\ell \in \{1, \ldots, L_{\text{tr}}\}$ and head $h \in \{1, \ldots, H\}$, we compute

$$\mathbf{Q}_{\ell,h}^{(s)} = \mathbf{Z}^{(s,\ell-1)}\mathbf{W}_{Q,\ell,h}, \qquad (12)$$
$$\mathbf{K}_{\ell,h}^{(s)} = \mathbf{Z}^{(s,\ell-1)}\mathbf{W}_{K,\ell,h}, \qquad (13)$$
$$\mathbf{V}_{\ell,h}^{(s)} = \mathbf{Z}^{(s,\ell-1)}\mathbf{W}_{V,\ell,h}, \qquad (14)$$

where $\mathbf{W}_{Q,\ell,h}, \mathbf{W}_{K,\ell,h}, \mathbf{W}_{V,\ell,h} \in \mathbb{R}^{d \times d_h}$. We apply affine modulation to the attention logits:

$$\gamma_{\ell,h}^{(s)}, \ \beta_{\ell,h}^{(s)} = g_{\ell,h}\left(z_{\text{pheno}}^{(s)}\right), \tag{15}$$

$$\mathbf{A}_{\ell,h}^{(s)} = \text{softmax}\left(\gamma_{\ell,h}^{(s)} \cdot \frac{Q_{\ell,h}^{(s)}\left(K_{\ell,h}^{(s)}\right)^{\top}}{\sqrt{d_h}} + \beta_{\ell,h}^{(s)}\right).$$

The head output is $\mathbf{O}_{\ell,h}^{(s)} = \mathbf{A}_{\ell,h}^{(s)} \mathbf{V}_{\ell,h}^{(s)}$, followed by the standard Transformer update. Finally, after $L_{\text{tr}}$ layers, we take the ROI token block from $Z^{(s,L_{\text{tr}})}$ as $\mathbf{X}_{\text{out}}^{(s)} \in \mathbb{R}^{N \times d}$. A permutation-invariant readout produces a brain-network embedding:

$$\mathbf{z}_{\text{brain}}^{(s)} = \text{Readout}\left(\mathbf{X}_{\text{out}}^{(s)}\right) \in \mathbb{R}^{d_b}. \tag{16}$$

### 2.5. Multi-modal Fusion

While phenotypes already condition routing and attention, we additionally fuse modality-level embeddings for final prediction. We first project $\mathbf{z}_{\text{tab}}^{(s)}$ and $\mathbf{z}_{\text{text}}^{(s)}$ to the same dimension as $\mathbf{z}_{\text{brain}}^{(s)}$:

$$\bar{\mathbf{z}}_{\text{tab}}^{(s)} = \mathbf{W}_{\text{tab}}\mathbf{z}_{\text{tab}}^{(s)} \in \mathbb{R}^{d_b}, \quad \bar{\mathbf{z}}_{\text{text}}^{(s)} = \mathbf{W}_{\text{text}}\mathbf{z}_{\text{text}}^{(s)} \in \mathbb{R}^{d_b}, \tag{17}$$

where $\mathbf{W}_{\text{tab}} \in \mathbb{R}^{d_b \times d_p}$ and $\mathbf{W}_{\text{text}} \in \mathbb{R}^{d_b \times d_\ell}$. We then compute gating weights

$$g^{(s)} = \text{softmax}\left(\mathbf{W}_g[\mathbf{z}_{\text{brain}}^{(s)}\|\bar{\mathbf{z}}_{\text{tab}}^{(s)}\|\bar{\mathbf{z}}_{\text{text}}^{(s)}]\right) \in \mathbb{R}^3, \tag{18}$$

where $\mathbf{W}_g \in \mathbb{R}^{3 \times 3d_b}$. The fused embedding is

$$\mathbf{z}_{\text{fuse}}^{(s)} = g_1^{(s)}\mathbf{z}_{\text{brain}}^{(s)} + g_2^{(s)}\bar{\mathbf{z}}_{\text{tab}}^{(s)} + g_3^{(s)}\bar{\mathbf{z}}_{\text{text}}^{(s)} \in \mathbb{R}^{d_b}. \tag{19}$$

Finally, a prediction head yields $\hat{y}^{(s)} = h(\mathbf{z}_{\text{fuse}}^{(s)})$.

### 2.6. Training Process

Given a mini-batch $\mathcal{B}$ of size $B$, we optimize a supervised loss $\mathcal{L}_{\text{sup}}$:

$$\mathcal{L}_{\text{sup}} = -\frac{1}{B}\sum_{s \in \mathcal{B}}\sum_c y_c^{(s)}\log\left(\text{softmax}(\hat{y}^{(s)})_c\right). \tag{20}$$

Furthermore, to encourage brain and phenotype feature consistency and reduce shortcut reliance, we add an InfoNCE alignment loss between $\mathbf{z}_{\text{brain}}^{(s)}$ and $\mathbf{z}_{\text{pheno}}^{(s)}$. Since $\mathbf{z}_{\text{brain}}^{(s)} \in \mathbb{R}^{d_b}$ and $\mathbf{z}_{\text{pheno}}^{(s)} \in \mathbb{R}^{d_c}$ may have different dimensions, we use linear projection heads:

$$\mathbf{u}^{(s)} = \mathbf{W}_b\mathbf{z}_{\text{brain}}^{(s)} \in \mathbb{R}^{d_a}, \qquad \mathbf{v}^{(s)} = \mathbf{W}_p\mathbf{z}_{\text{pheno}}^{(s)} \in \mathbb{R}^{d_a}, \tag{21}$$

where $\mathbf{W}_b \in \mathbb{R}^{d_a \times d_b}$ and $\mathbf{W}_p \in \mathbb{R}^{d_a \times d_c}$. We define cosine similarity $\text{sim}(\mathbf{u}, \mathbf{v}) = \frac{\mathbf{u}^{\top}\mathbf{v}}{\|\mathbf{u}\| \|\mathbf{v}\|}$ and compute

$$\mathcal{L}_{\text{align}} = -\frac{1}{B}\sum_{s \in \mathcal{B}}\log\frac{\exp\left(\text{sim}(\mathbf{u}^{(s)}, \mathbf{v}^{(s)})/\tau\right)}{\sum_{s' \in \mathcal{B}}\exp\left(\text{sim}(\mathbf{u}^{(s)}, \mathbf{v}^{(s')})/\tau\right)}, \tag{22}$$

where $\tau$ is a temperature. Therefore, the final objective is:

$$\mathcal{L} = \mathcal{L}_{\text{sup}} + \lambda\,\mathcal{L}_{\text{align}}, \tag{23}$$

Where $\lambda$ is a hyper-parameter.

## 3. Experiments

In this section, we conduct a multifaceted evaluation of PhenoBrain to demonstrate its effectiveness. Furthermore, we validate its long-range communication mechanism and the effectiveness of its various modalities and components. Finally, we assess the framework's biological interpretability by validating whether its learned routing patterns and attention mechanisms can generate stable disease prediction biomarkers with neuroscientific significance.

### 3.1. Experiment Settings

In this paper, we evaluate the performance of PhenoBrain against two representative benchmarks: ABIDE and PIXAR. Specifically, functional connections are constructed as graphs using Pearson correlation coefficients and sparse thresholding. Phenotypic modalities include structured clinical variables (such as age, site, and scale scores) and unstructured natural language descriptions constructed by domain experts. We compare our model with nine state-of-the-art (SOTA) benchmarks covering classic GNNs and state-of-the-art phenotypic/cue-driven methods, using metrics including ACC, AUC, SEN, and SPE. All experiments were performed on an NVIDIA RTX 5090 GPU using the Adam optimizer, with standard regularization and missing value imputation applied. For complete technical details regarding dataset partitioning, benchmark model architecture, hyperparameter configuration, and data preprocessing, please refer to Appendix A.2.

### 3.2. Superiority Analysis

#### 3.2.1. MAIN PERFORMANCE ANALYSIS

Table 1 summarize the 4 main evaluation metrics results on ABIDE and PIXAR. Overall, the proposed PhenoBrain achieves state-of-the-art performance against all compared baselines. Specifically, on the ABIDE dataset, PhenoBrain achieved an accuracy of 0.7635 and an AUC of 0.8346, representing performance improvements of 3.40% and 3.46% respectively compared to the top-performing baseline model MGDR. Furthermore, PhenoBrain exhibits significantly im-

*Table 1.* Performance comparison of our PhenoBrain with latest multi-modal brain network analysis baseline models on two constructed datasets.

| Methods | Dataset: ABIDE | | | | Dataset: PIXAR | | | |
|---|---|---|---|---|---|---|---|---|
| | ACC | AUC | SEN | SPE | ACC | AUC | SEN | SPE |
| InceptionGCN | 0.5731± 0.0374 | 0.6323± 0.0256 | 0.7049± 0.1785 | 0.5874± 0.097 | 0.7887± 0.1223 | 0.8031± 0.0940 | 0.7699± 0.1005 | 0.8249± 0.1632 |
| PopGCN | 0.6035± 0.075 | 0.6545± 0.083 | 0.7300± 0.1040 | 0.6121± 0.0733 | 0.7988± 0.0512 | 0.8061± 0.0433 | 0.7767± 0.0689 | 0.8230± 0.0877 |
| LSTMGCN | 0.6418± 0.1434 | 0.7137± 0.1098 | 0.7543± 0.1696 | 0.6333± 0.0731 | 0.8257± 0.1228 | 0.8180± 0.1645 | 0.7973± 0.1298 | 0.8396± 0.1375 |
| EV-GCN | 0.6842± 0.0662 | 0.7246± 0.0834 | 0.7917± 0.0877 | 0.6616± 0.0451 | 0.8399± 0.0713 | 0.8415± 0.0615 | 0.8220± 0.0553 | 0.8534± 0.0898 |
| Multi-GCN | 0.6669± 0.1118 | 0.7288± 0.1342 | 0.7715± 0.0865 | 0.6422± 0.0641 | 0.8450± 0.1439 | 0.8538± 0.1230 | 0.8313± 0.1276 | 0.8445± 0.0854 |
| LGL | 0.7113± 0.0886 | 0.7732± 0.1207 | 0.8041± 0.1096 | 0.6078± 0.0655 | 0.8778± 0.0932 | 0.8891± 0.1477 | 0.8605± 0.0729 | 0.8655± 0.0842 |
| MMGL | 0.6746± 0.0944 | 0.7233± 0.1179 | 0.6879± 0.1325 | 0.6774± 0.0918 | 0.8882± 0.0431 | 0.9087± 0.0720 | 0.8676± 0.0930 | 0.8448± 0.1030 |
| SPromptGL | 0.7084± 0.0788 | 0.7564± 0.0914 | 0.8289± 0.1131 | 0.6731± 0.0618 | 0.8780± 0.1289 | 0.8977± 0.1334 | 0.9007± 0.0745 | 0.8753± 0.0619 |
| MGDR | 0.7384± 0.0531 | 0.8067± 0.0512 | **0.8864± 0.0436** | 0.6533± 0.0798 | 0.9026± 0.0563 | 0.9163± 0.1032 | 0.8813± 0.0712 | 0.8766± 0.0334 |
| **PhenoBrain** | **0.7635± 0.0272** | **0.8346± 0.0498** | 0.8384± 0.0317 | **0.6923± 0.0487** | **0.9167± 0.0517** | **0.9259± 0.0644** | **0.9333± 0.0411** | **0.8889± 0.0512** |

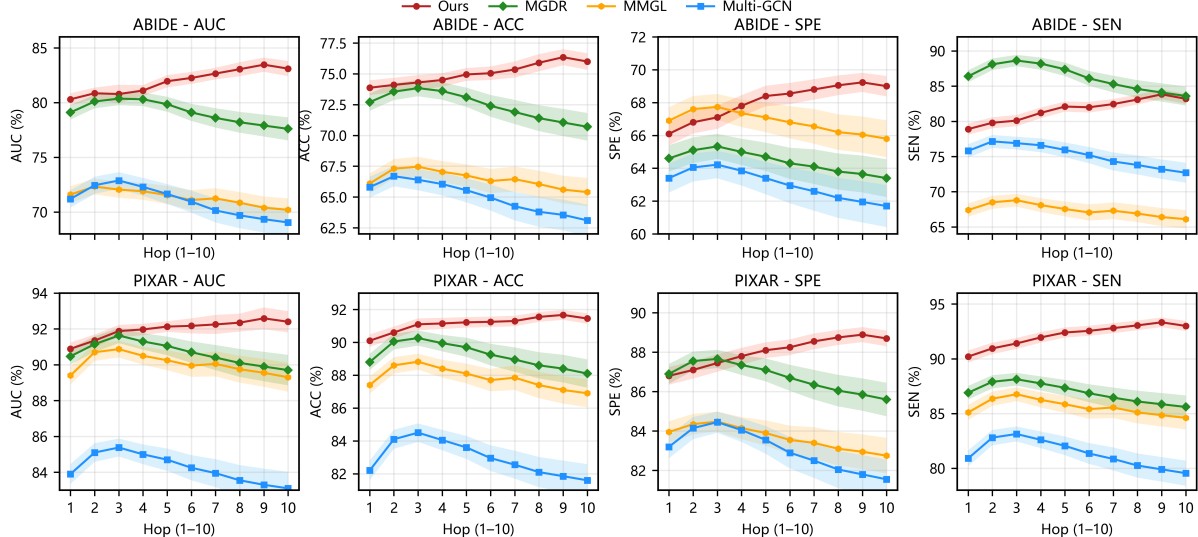

*Figure 3.* Performance comparison of PhenoBrain and 3 latest baseline models at different distance lengths.

proved specificity compared to MGDR, suggesting that phenotypic conditionation mechanisms help the model reduce false positives and learn a more balanced decision boundary. For the PIXAR dataset, PhenoBrain also yields satisfied improvement, which ACC and AUC reach 0.9167 and 0.9259. Compared with the strongest baseline MGDR, PhenoBrain improves ACC by relative 1.56% and AUC by relative 1.05%. Moreover, PhenoBrain achieves best SEN, surpassing the best baseline (SPromptGL) by 3.62%. These results suggest that PhenoBrain generalizes beyond ASD and benefits disease prediction in a different clinical setting.

### 3.2.2. LONG-RANGE COMMUNICATION ANALYSIS

To demonstrate PhenoBrain's advantages in long-range communication, we compared the experimental results of three state-of-the-art methods on two datasets under different hop counts, which is shown in Figure 3. Experimental results show that our method remains competitive on both long and short distances, outperforming state-of-the-art baseline methods on most metrics across both datasets. This demonstrates that the proposed routing method does not sacrifice lo-

cal information. In general, for short-range communication, our method offers limited improvement on both datasets, as existing GCN-based methods already capture short-range dependencies well. However, for long-range communication, PhenoBrain shows significant and stable performance with increasing distance on most metrics. Specifically, on the ABIDE dataset, the baseline methods MMGL, Multi-GCN, and MGDR achieve optimal performance with 2, 3, and 3 hops, respectively. On the PIXAR dataset, all three methods reach their optimal performance with 3 hops, but performance gradually decreases with increasing hops due to oversmoothing. Our method, however, shows a stable performance increase with increasing hop count. Furthermore, we found that PhenoBrain performs worse than the baseline MGDR on the ABIDE dataset in terms of SEN (Search Estimation), because MGDR tends to select high-recall decision boundaries for positive classes, such as stronger positive class aggregation. Nevertheless, our method still steadily improves performance on this metric with increasing hop count. Experimental results show that our method remains competitive in both long-range and short-range domains.

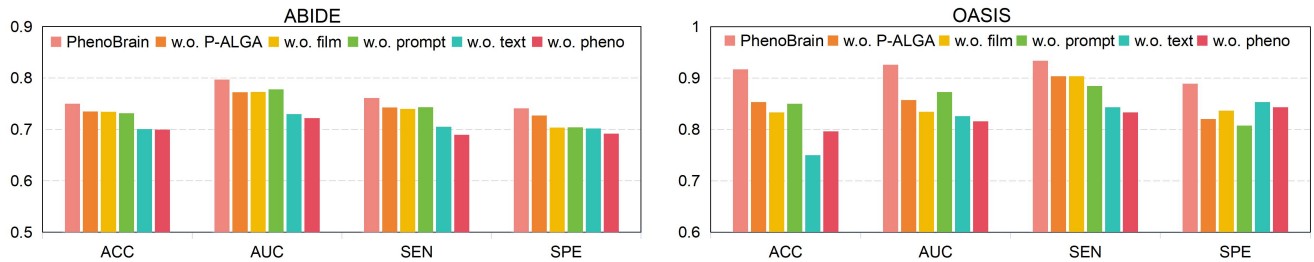

*Figure 4.* Ablation analysis result of each modals and components of our PhenoBrain.

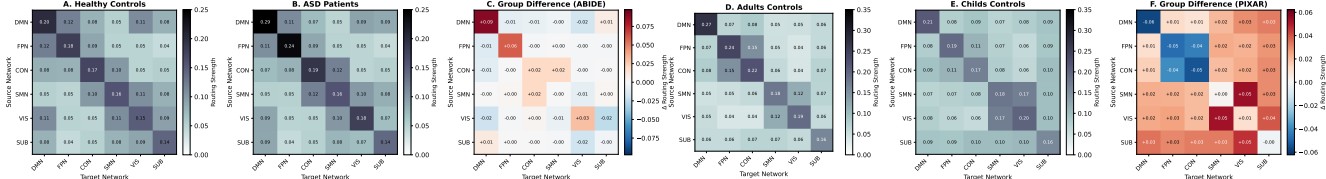

*Figure 5.* Panels A-C (ABIDE dataset) and Panel D-F (PIXAR dataset) illustrate long-range routing patterns in the brains of different groups, where the matrix values represent the intensity of neural signal propagation between two brain regions. Specifically, Panels A, B, and C show the long-range routing matrices and differences between healthy brains and brains of patients with ASD, respectively. Panels D, E, and F show the long-range routing matrices and differences between adult brains and children's brains, respectively

## 3.3. Ablation Study

In order to show the effectiveness of each components of PhenoBrain, we conduct ablation studies to validate the contribution of each key design in PhenoBrain, including (i) removing phenotype-conditioned long-range routing (w/o PCLR); (ii) removing phenotype modulation on attention logits (w/o FiLM); (iii) removing phenotype prompts (w/o prompt); (iv) removing textual phenotype (w/o text); and (v) removing phenotype conditioning (**w/o pheno**). Experimental results are shown in Figure 4, demonstrating that each component of the framework is meaningful. First, compared to the full model, removing the textual and phenotypic modalities reduced the ACC by 7.02% and 7.21% on the ABIDE dataset, respectively, and by 22.2% and 15.1% on the PIXAR dataset, respectively. This indicates that multimodal phenotypic narratives provide supplementary information beyond structured variables, and that prompts can serve as effective semantic probes. Furthermore, removing the PCLR module, attention modulation, and prompt tags reduced the model's ACC by 2.04%, 2.17%, and 2.58% on the ABIDE dataset, respectively, and by 7.85%, 10%, and 7.43% on the PIXAR dataset, respectively. This confirms that explicitly modeling long-range communication of phenotypic conditioning can provide important predictive signals and serve as an interpretability component.

## 3.4. Interpretability Analysis

Beyond predictive performance, we also analyzed whether PhenoBrain provides meaningful and stable interpretations through its outputs at two levels: phenotypic conditional routing and attention to ROIs. Due to space limitations, a complete analysis can be found in Appendix A.3.

Firstly, Figure 5 illustrates and explains the long-range communication routing patterns learned by our model, as well as the differences between different populations. Our core argument is that the model should not only output a black-box embedding, but should also output interpretable evidence of brain network mechanisms. Therefore, we visualize the long-range routing strength learned by PCLR as a network-level matrix, where the matrix values represent the intensity of neural signal propagation between two brain regions. Specifically, for each subject, a long-range routing matrix was first obtained from PCLR, which modeled the strength and reachability of multi-hop propagation, and then ROIs were grouped according to six standard functional networks. The vertical axis represents the Source Network, and the horizontal axis represents the Target Network. The results of Panels A–C show that the "cross-network communication patterns" The difference between ASD patients and healthy individuals is reflected in the strength differences related to the differentially expressed neural network (DMN) and the inter-brain neural network (FPN). Compared to the normal control group, ASD patients exhibited overactive DMN networks and weakened cross-network connectivity, indicating FPN abnormalities. Panels D–E showed that adults had stronger intra- and intra-network connections (CON), indicating a more mature control and memory system. In contrast, children's sensorimotor system showed stronger internal and inter-network connectivity (VIS and SMN), but weaker FPN and CON.

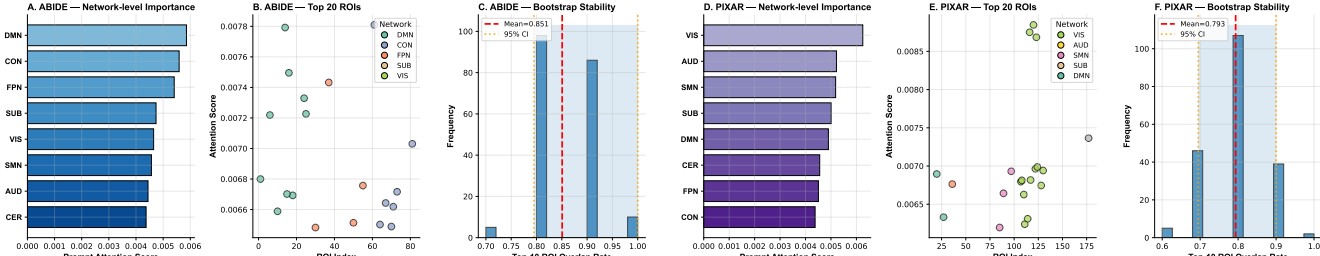

*Figure 6.* Figures A-C (ABIDE dataset) and D-F (PIXAR dataset) use the same ROI-to-network assignment; that is, the 200 ROIs are divided into eight typical networks for direct comparison. Figures A and D show the network-level importance ranking in both datasets. Figures B and E show the importance at the ROI level, visualizing the top-20 ROIs ranked by average attention. Figures C and F show the stability of the results, displaying the overlap distribution of the top-10 ROIs across 200 resampling iterations using histograms.

Furthermore, Figure 6 demonstrates the evidence from the locate brain network in text and phenotypic formation, and this interpretation remains stable under resampling. Specifically, we derive the prompt-to-ROI attention weights from the Transformer and summarize them into the average Prompt Attention Score of each ROI. Based on this, we perform visualization on three levels, including: (1) Network-level importance (Panel A, D): We calculate the mean of the attention scores of the ROIs according to their respective networks and sort them to obtain the ranking of the importance of the functional networks; (2) ROI-level importance (Panel B, E): We sort all ROIs according to their average attention scores and display the top-20 ROIs; the horizontal axis is the ROI Index and the vertical axis is the Attention Score. Each point represents an ROI, thus showing the spatial distribution of key ROIs; and (3) Bootstrap stability (Panel C, F): We perform 200 resampling iterations, recalculate the top-10 ROI list each time, and count the overlap rate of the top-10 ROIs, and display their distribution in a histogram. Experimental results in Panels A and D show that in the ABIDE dataset, networks controlling behavior and social interaction are relatively more important, while in the PIXAR, networks controlling perception are more important. Meanwhile, the ROI-level distribution results in Panels B and E also correspond to the network-level results.

## 4. Conclusion

We propose a phenotype-conditional multimodal brain network analysis framework, termed PhenoBrain, which uses structured phenotype data and multi-modal data such as natural language descriptions as context to represent individual differences, guiding the model to transmit information and select evidence in the brain network. Specifically, we propose a phenotype-conditional long-range router, constructing a multi-hop routing kernel on the brain map and generating individual-specific long-range communication representations with low-rank, phenotype-related modulation, thereby achieving pathway-level interpretability. Furthermore, we construct two phenotype-based multi-modal

brain network analysis datasets based on existing image data, demonstrating the superiority and robustness of our proposed method through extensive and thorough experiments.

## Impact Statement

This paper presents work whose goal is to advance the field of Machine Learning. There are many potential societal consequences of our work, none which we feel must be specifically highlighted here.

## Acknoledgement

The research is supported by the National Natural Science Foundation of China (no. 62325604, 62276271, 62441618, 62506371), the Major Program Project of Xiangjiang Laboratory (no. 24XJJCYJ01002), and the Hunan Province Graduate Student Innovation Project (no. XJJC2025013).

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

# A. Appendix

## A.1. Related Work

### A.1.1. MULTI-IMAGE MULTI-MODAL BRAIN NETWORK ANALYSIS

Multi-image multi-modal brain network analysis has been a critical area of research in neuroimaging, where different imaging modalities, such as fMRI, EEG, and structural MRI, are integrated to study brain networks. Several methods have explored the fusion of multi-modal imaging data to improve the understanding of brain connectivity and its relationship to various disorders. MMGL (Zheng et al., 2022) is a method that employs multi-view learning to integrate multiple brain graphs obtained from different imaging modalities. MMGL focuses on clustering brain networks by learning a shared latent space across views, aiming to uncover coherent brain regions that correlate with specific functional states or disease conditions. CroGen (Luo et al., 2022) takes a similar approach, but it also generates missing network information across views. By filling in missing data, CroGen can better handle incomplete multi-modal datasets, improving the accuracy of brain network analysis for disorder diagnosis. RTGNN (Zhao et al., 2022) extends graph neural networks by incorporating tensor-based multi-view data. RTGNN learns the joint representation of multi-modal brain networks through reinforced aggregation techniques, enhancing the model's ability to capture long-range dependencies and multi-scale patterns across different modalities. Cross-GNN (Yang et al., 2023) uses a cross-modal mutual learning approach to map multi-modal brain connectomes, where the model learns to align representations from different modalities to improve classification performance for brain disorders. This method shows significant improvement in diagnosis accuracy by learning a joint representation of structural and functional connectivity. While these methods focus on integrating multi-modal imaging data for brain network analysis, they largely assume that the imaging modalities are the primary sources of information. This overlooks the potential of integrating phenotypic data, such as clinical and behavioral characteristics, which are crucial for understanding individualized brain network patterns and diseases.

### A.1.2. PHENOTYPIC-DRIVEN MULTI-MODAL BRAIN NETWORK ANALYSIS

Phenotypic-driven multi-modal brain network analysis, which incorporates clinical and behavioral information alongside neuroimaging data, is an emerging research area. These approaches aim to better personalize brain network models by accounting for the complex interplay between brain structure, function, and the individual's phenotype. MGDR (Jiang et al., 2024) introduces a disentangled representation learning framework that separates different sources of information, such as imaging and phenotypic data. By learning distinct representations for different modalities, MGDR can better predict brain diseases, improving interpretability and predictive power. This method focuses on making the network predictions more robust by disentangling the contributions of different data sources. SPromptGL (Wan et al., 2025) takes a novel approach by using semantic prompts to guide graph learning. This method leverages textual phenotype descriptions as prompts to help the model focus on relevant brain regions and connectivity patterns. By incorporating semantic prompts into graph learning, SPromptGL improves the model's ability to capture phenotype-related variations in brain network structures, aiding in the diagnosis of brain diseases. Despite the promise of these methods, they still face limitations in fully integrating phenotypic data into the brain network learning process. Current approaches often treat phenotypic data as a separate input during the late fusion stage or use simplistic models that do not fully capture the complex relationships between phenotype and brain connectivity. Our approach overcomes these limitations by introducing a mechanism-level fusion framework, which allows phenotypic information to directly influence how brain network representations are learned, leading to more individualized and interpretable brain network models.

## A.2. Experiments

### A.2.1. USED DATASETS

We evaluate PhenoBrain on two representative multi-modal brain network analysis benchmarks that provide fMRI-derived connectomes together with phenotype information. Specifically, we report classification results on ABIDE (Craddock et al., 2013) (ASD vs. healthy control), following the common practice of connectome-based disease prediction. In addition, we include PIXAR (children vs. adults) to study phenotype-dependent routing patterns and prompt-based explanations under a developmental setting. For each subject, we construct a functional connectome from ROI time series using Pearson correlation, and obtain a sparse graph by thresholding absolute FC values. Furthermore, the phenotype modality contains (i) structured variables (e.g., age/sex/site/clinical scores when available) and (ii) natural language textual descriptions, which are encoded as described in Section 2.3.

### A.2.2. COMPARED BASELINES

We compare PhenoBrain with 9 strong connectome learning baselines, covering classical supervised graph models and recent phenotype-based multi-modal / prompt-based methods, including InceptionGCN (Kazi et al., 2019a), PopGC (Parisot et al., 2017), LSTMGCN (Kazi et al., 2019b), EV-GCN (Huang & Chung, 2020), Multi-GCN (Kazi et al., 2019c), LGL (Cosmo et al., 2020), MMGL (Zheng et al., 2022), SPromptGL (Wan et al., 2025), MGDR (Jiang et al., 2024). All methods are evaluated under the same metrics: Accuracy (ACC), AUC, Sensitivity (SEN), and Specificity (SPE). The details of the baseline comparison can be found in the appendix.

### A.2.3. IMPLEMENTATION DETAILS

All experiments are implemented using PyTorch and conducted on a single NVIDIA RTX 5090 GPU. For each dataset, we compute FC matrices and build sparse adjacency matrices by thresholding.For text and tabular phenotypes, structured phenotypes are z-score normalized, and missing values are imputed with missingness indicators when needed, while textual phenotypes are tokenized and truncated/padded to a fixed length. Furthermore, we train all models end-to-end with Adam and apply standard regularization.

### A.2.4. INTERPRETABILITY ANALYSIS

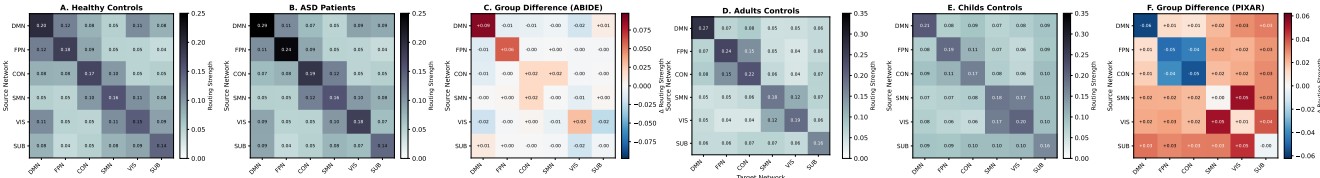

*Figure 7.* Panels A-C (ABIDE dataset) and Panel D-F (PIXAR dataset) illustrate long-range routing patterns in the brains of different groups, where the matrix values represent the intensity of neural signal propagation between two brain regions. Specifically, Panels A, B, and C show the long-range routing matrices and differences between healthy brains and brains of patients with ASD, respectively. Panels D, E, and F show the long-range routing matrices and differences between adult brains and children's brains, respectively

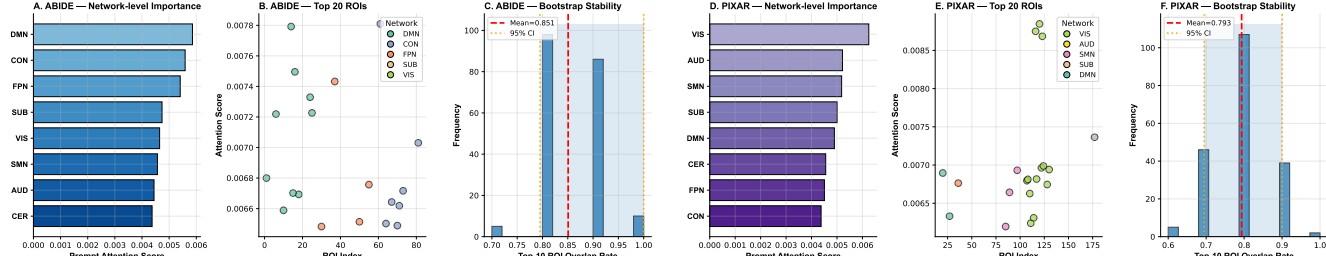

*Figure 8.* Figures A-C (ABIDE dataset) and D-F (PIXAR dataset) use the same ROI-to-network assignment; that is, the 200 ROIs are divided into eight typical networks for direct comparison. Figures A and D show the network-level importance ranking in both datasets. Figures B and E show the importance at the ROI level, visualizing the top-20 ROIs ranked by average attention. Figures C and F show the stability of the results, displaying the overlap distribution of the top-10 ROIs across 200 resampling iterations using histograms.

Beyond predictive performance, we also analyzed whether PhenoBrain provides meaningful and stable interpretations through its outputs at two mechanism levels: phenotypic conditional routing and attention to ROIs.

Firstly, Figure 7 illustrates and explains the "long-range communication routing patterns" learned by our model, as well as the differences between different populations. Our core argument is that the model should not only output a black-box embedding, but should also output interpretable evidence of brain network mechanisms. Therefore, we visualize the long-range routing strength learned by PCLR as a network-level matrix: Panels A–C correspond to ABIDE (Healthy Controls, ASD Patients, and the differences between the two), and Panels D–F correspond to PIXAR (Adults, Children, and the differences between the two), where the matrix values represent the intensity of neural signal propagation between two brain regions. Specifically, for each subject, a long-range routing matrix was first obtained from PCLR, which modeled the strength and reachability of multi-hop propagation, and then ROIs were grouped according to six standard functional networks. The vertical axis represents the Source Network, and the horizontal axis represents the Target Network. The

results of Panels A–C show that the "cross-network communication patterns" The difference between ASD patients and healthy individuals is reflected in the strength differences related to the differentially expressed neural network (DMN) and the inter-brain neural network (FPN). The DMN and FPN networks control social cognition and attentional memory functions, respectively. Compared to the normal control group, ASD patients exhibited overactive DMN networks and weakened cross-network connectivity, indicating FPN abnormalities. Panels D–E showed that adults had stronger intra- and intra-network connections (CON), indicating a more mature control and memory system. In contrast, children's sensorimotor system showed stronger internal and inter-network connectivity (VIS and SMN), but weaker FPN and CON. Traditional graph learning methods struggle to capture long-range correlations across brain regions, challenging the addresses of our proposed PhenoBrain.

Furthermore, Figure 8 demonstrates the evidence from the "query/locate" brain network in text and phenotypic formation, and this interpretation remains stable under resampling. Specifically, on the ABIDE and PIXAR datasets, we derive the prompt-to-ROI attention weights from the Transformer and summarize them into the average Prompt Attention Score of each ROI. Based on this, we perform visualization on three levels, including: (1) Network-level importance (Panel A, D): We calculate the mean of the attention scores of the ROIs according to their respective networks and sort them to obtain the ranking of the importance of the functional networks; (2) ROI-level importance (Panel B, E): We sort all ROIs according to their average attention scores and display the top-20 ROIs; the horizontal axis is the ROI Index and the vertical axis is the Attention Score. Each point represents an ROI, thus showing the spatial distribution of key ROIs; and (3) Bootstrap stability (Panel C, F): We perform 200 resampling iterations, recalculate the top-10 ROI list each time, and count the overlap rate of the top-10 ROIs, and display their distribution in a histogram. Experimental results in Panels A and D show that in the ABIDE dataset, networks controlling behavior and social interaction, such as DMN, CON, and FPN, are relatively more important, while in the PIXAR dataset, networks controlling perception, such as VIS and AUD, are more important. Meanwhile, the ROI-level distribution results in Panels B and E also correspond to the network-level results. The conclusions of the two visualization analyses are largely consistent with the experimental results shown in the long-range routing pattern of Figure 8, demonstrating that the proposed PhenoBrain can provide evidence for brain network analysis at both the network and ROI levels. Furthermore, the stability experimental results shown in Panels C and F also demonstrate the reliability of the visualization results.

