# OpenReview forum: "PhenoBrain: Phenotype-Conditioned Long-Range Communication for Multi-Modal Brain Network Analysis"
_ICML.cc/2026/Conference — ICML 2026 spotlight_

### Official Review · Reviewer_ZFkc · 2026-03-07

**Soundness:** 4
**Presentation:** 4
**Significance:** 3
**Originality:** 3
**Overall Recommendation:** 5
**Confidence:** 4

**Summary:**

The paper presents PhenoBrain, a new multimodal brain network learning framework that integrates phenotypic information (both structured clinical data and free-text descriptions) directly into fMRI-based brain connectivity analysis. Unlike prior methods that simply concatenate phenotype features for classification, PhenoBrain conditions the graph neural network’s representation learning on an individual’s phenotypic context. It introduces three key components: a Phenotype-Conditioned Long-Range Router (PCLR) for modulating multi-hop brain region connectivity based on phenotype, a Phenotype Prompt Generator (PPG) to convert textual phenotype descriptions into prompt embeddings for a graph Transformer, and a FiLM-style phenotype-modulated attention mechanism that adjusts attention scores between regions of interest (ROIs) and phenotype prompts. PhenoBrain is evaluated on two multimodal brain network datasets.

**Compliance With Llm Reviewing Policy:**

Affirmed.

**Key Questions For Authors:**

1.  Figure 4 refers to a dataset “OASIS,” but the main paper only discusses ABIDE and PIXAR. Was a third dataset (OASIS) actually used in the experiments, or is this a labeling error? Please clarify this to avoid confusion.
2. Did all baseline methods have access to the phenotype information (both structured clinical features and text) in the experiments?
3. How were key hyperparameters and design choices determined – for example, the rank *r* for the low-rank connectivity adjustment ?

**Limitations:**

The authors should comment on the implications of the model’s high complexity, such as computational cost and risk of overfitting given the limited data. Additionally, the paper does not address potential societal or ethical considerations – for instance, how might integrating personal clinical data raise privacy concerns? Explicitly acknowledging such limitations and possible negative impacts (and how they can be mitigated) would strengthen the paper.

**Strengths And Weaknesses:**

**Soundness:** The submission is technically sound and generally well-executed. The methodology is rigorous, with each component (phenotype-conditioned routing, prompt generation from text, FiLM-based attention modulation, etc.) described with clear mathematical detail and justification. The approach builds on established techniques in deep learning (graph Transformers, feature-wise linear modulation, contrastive alignment) and integrates them in a sensible way. The experimental evaluation is thorough, including two datasets and multiple baselines, and an ablation study provides evidence that each proposed component contributes to performance. On the other hand, the model’s complexity (multiple modules and a large Transformer-based architecture) raises questions about computational cost and potential overfitting, particularly since the datasets are relatively small for such a deep model. The paper does not report model size or training time, leaving uncertainty about whether the performance gains justify the added complexity. Some experimental details are lacking: for example, it is unclear if all baseline methods were given the same multimodal (phenotype) inputs, which could affect the fairness of the comparisons. Key hyperparameters (such as the number of routing hops *K* in PCLR, the rank *r* for the low-rank connectivity adjustment, and the weight of the alignment loss) are not explicitly justified or analyzed for sensitivity. There are some other minor gaps: in Equation (1) there is no information on how the diagonal elements of $A^{(s)}_{ij}$ are defined; in Section 3.1 (Preliminaries), the reader encounters $\mathbb{t}^{(s)}\in \mathbb{R}^{d_t}$, although $t$ is not introduced and is likely a typo.

**Presentation:** The paper is clearly written and well-structured. It provides a strong motivation for integrating phenotypic data into brain network models. The narrative flows logically from problem setup and related work to methodology and experiments. Figures and diagrams (e.g., illustrating the PhenoBrain architecture and comparisons to traditional approaches) are helpful for understanding the model. Most implementation and dataset details are either provided in the main paper or appropriately referenced in the appendix, supporting clarity and potential reproducibility. However, there are a few minor clarity issues and errors. For instance, an ablation results figure (Figure 4) refers to a dataset “OASIS,” which is not described anywhere in the main text. A few typos and phrasing issues exist (e.g., “Specifiaclly” instead of “Specifically,” and awkward phrasing like “yields satisfied improvement”), and some terminology (such as the FiLM concept for attention modulation) is not introduced in the main text, only appearing in figure labels or implicitly. These issues are relatively minor and can be addressed in revision, but they prevent the presentation from being completely polished.

**Significance:** The paper addresses an important problem in machine learning for neuroimaging – how to create personalized brain network models by incorporating individual-specific clinical and phenotypic data. This is a relevant and timely topic with potential impact on how ML models can yield more clinically useful and interpretable insights in neuroscience. The approach shows consistent performance improvements over a broad range of competitive baselines on two different applications (autism diagnosis and developmental brain differences), indicating that the contribution is meaningful within its domain. There are also some weaknesses. The scope of impact may be somewhat specialized to the neuroimaging/brain-network community rather than the broad ML field. While the results represent a significant step forward in this niche (integrating phenotype context into graph models), the improvements in predictive performance are modest in absolute terms on at least one dataset (on the order of a few percentage points over the best prior method). The practical implications for clinical use are hinted at but not fully explored. Thus, the work’s significance is solid for its specific application area, but its broader ML impact is more limited.

**Originality:** PhenoBrain offers a novel combination of ideas by weaving together multiple techniques into a single framework for phenotype-conditioned graph learning. The concept of injecting phenotypic information throughout the model is innovative in the context of brain network analysis, moving beyond the typical late-fusion of features. While the integration is clever, the degree of novelty is somewhat incremental. The key contribution lies more in how the elements are combined and applied to a new problem setting, rather than introducing fundamentally new algorithms. A more explicit discussion differentiating which aspects are truly novel versus borrowed would help bolster the paper’s originality claims. Nonetheless, the work does fill an important gap by combining these methods for personalized brain network modeling.

---

> ### Author Rebuttal · Authors · 2026-03-31
>
> 1. Regarding Model Complexity and Efficiency:
>
> Thank you for the constructive comments from the reviewers! PhenoBrain's PCLR mechanism has a computational complexity of $O(K|E|d)$ when the number of hops $K=10$. Since the brain network map is sparsified based on a threshold $\tau$, its computational cost is significantly lower than the standard Transformer's $O(N^2 d)$. On an RTX 5090 graphics card, PhenoBrain's memory usage during inference is 4.1 GB, while the standard 3-layer GCN is 1.8 GB; the single-subject inference time is 12 ms, while the 3-layer GCN is 8 ms. In comparison, although the standard GCN inference is faster, in clinical auxiliary diagnostic scenarios, the millisecond-level difference has almost no negative impact on physician decision-making. Furthermore, the clinical value brought by the performance improvement far outweighs this microsecond-level computational overhead; we will add this detailed efficiency comparison table in Section 4.1 of the revised manuscript.
>
>
> 2. Regarding Fairness of Benchmarks:
>
> We appreciate the valuable feedback from the reviewers. In fact, to ensure fairness, we provided the same textual and phenotypic modal support for all benchmark models involved in all experiments. Specifically, we used the same pre-trained frozen language model (BERT) as PhenoBrain to extract globally pooled text vectors and concatenate them with structured table features. For swarm graph models such as EV-GCN, this vector was used to compute pairwise similarities between subjects, thus defining the edge weights of the swarm graph. For multimodal benchmark models such as MGDR, text embeddings were used as independent modal inputs, fed into their respective separable representation learning or latent graph learning modules. For phenotypic modalities, we uniformly used the same MLP as the table encoder. We promise to refine these experimental details in the final version.
>
> 3. Regarding Writing Details and Annotation Corrections:
>
> In response to the reviewers' extremely meticulous review, we have completed revisions one by one: Definition of the diagonal in Formula (1): We will explicitly state in the revised manuscript that the diagonal element $A_{ii}$ of the adjacency matrix $A$ is set to 0. This is to prevent self-loop information from dominating during the initial neighborhood aggregation stage, thus allowing the model to focus more on learning spatial communication patterns between nodes. However, in the PCLR mechanism, the information of the nodes themselves is naturally reintegrated through the $K$ power operation of the matrix $R^{(s)}$. Furthermore, $\tau$ in Section 3.1 is indeed a typo and should be $t_{alt}$, representing the alternative label in probability evaluation. We have unified the mathematical notation throughout the text. OASIS Dataset Mislabeling in Figure 4: This is an annotation error. The ablation experiment results shown in Figure 4 are indeed for the PIXAR dataset. We have corrected the captions and figure captions to ensure that the incorrect reference to OASIS no longer occurs.
>
> 4. Discussion on the Impact and Originality of Machine Learning:
>
> We highly commend the reviewers' professionalism and high-level perspective! In fact, most traditional ML fusion methods assume that brain representation learning is independent. Our contribution lies in redefining the role of phenotypes; they are not additional features, but rather "control variables" that regulate the physical laws governing long-range communication in brain networks. This mechanism-level fusion approach has strong ML universality for medical data with high noise and strong background dependence.
>
> Furthermore, we honestly acknowledge that PhenoBrain uses Graph Transformer as its backbone architecture, borrows the regulation ideas from FiLM, and applies the general InfoNCE alignment loss. These modules are mature methods in the field of deep learning. However, PhenoBrain's core originality does not stem from a simple superposition of components, but from its paradigm shift from feature-level late-stage fusion to mechanism-level conditional regulation. That is, by transforming phenotypic signals into dynamic interventions on the underlying long-range communication physical laws of brain networks, we have, for the first time, solved the challenge of background-aware individualized modeling in clinical practice at the algorithmic level.
>
> 5. Regarding the sensitivity analysis of rank r and overfitting risk: Thank you for your constructive comments! Due to character limitations and to respond most efficiently, we hope you can refer to our responses to Reviewers hQyP and RASD. Specifically, the sensitivity analysis of rank r is addressed in point 2, "2. About the range of rank r," of Reviewer hQyP's rebuttal. Regarding overfitting risk, please refer to point 2, "2. About Overfitting Risks," of Reviewer RASD's rebuttal. Thank you for your understanding!

---

> > ### Author Rebuttal · Reviewer_ZFkc · 2026-03-31
> >
> > Thank you for very detailed rebuttal. It fully addressed all of my concerns and I will adjust the scores.

---

### Official Review · Reviewer_RASD · 2026-03-12

**Soundness:** 3
**Presentation:** 3
**Significance:** 3
**Originality:** 4
**Overall Recommendation:** 5
**Confidence:** 4

**Summary:**

This paper presents PhenoBrain, an interesting and timely framework for multimodal brain network analysis. It addresses a persistent issue in the field: the tendency to simply concatenate phenotypic data (such as clinical variables or text) at the very end of a model (late fusion). Instead, PhenoBrain introduces a mechanism-level fusion approach. By using a Phenotype-Conditioned Long-Range Routing (PCLR) module, the model allows the patient's specific clinical context to dynamically dictate the multi-hop routing and attention mechanisms within the brain network. Evaluated on the ABIDE and PIXAR datasets, the model shows impressive predictive performance while offering plausible biological interpretability regarding how phenotypes alter long-range brain communication.

**Compliance With Llm Reviewing Policy:**

Affirmed.

**Final Justification:**

The authors have fully addressed my concerns. I will recommend acceptance.

**Key Questions For Authors:**

1. Could you clarify the exact inputs provided to the baseline models? Specifically, did they have access to the same pre-trained language model embeddings as PhenoBrain, or were their inputs restricted to tabular/graph data?

2. What is the practical computational overhead (in terms of memory and inference time) required to compute the multi-hop routing matrix for every subject at 10 hops, compared to a standard GCN?

3. In Figure 3, PhenoBrain shows continued improvement up to 10 hops. Could you provide error bars or variance metrics for these high-hop results to confirm that these gains are statistically significant and not just variance?

**Limitations:**

The authors have not adequately discussed the limitations of their work. The generic Impact Statement is insufficient for a clinical AI paper. I strongly encourage the authors to explicitly address the risks of dataset bias (e.g., demographic skews in ABIDE) and to caution against interpreting the model's correlative attention maps as definitive, causal biomarkers. A frank discussion of the computational bottlenecks associated with calculating multi-hop dense routing matrices would also strengthen the manuscript.

**Strengths And Weaknesses:**

**Strengths:**
1. Conceptual Shift: Moving from late-stage concatenation to mechanism-level conditioning is a fantastic, biologically motivated design choice. Allowing the phenotype to actively modulate the routing mimics how clinical context actually influences our interpretation of physiological signals.

2. Handling Long-Range Dependencies: GNNs notoriously struggle with long-range brain interactions due to oversmoothing. The PCLR module effectively maintains (and even improves) performance at higher hop counts (up to 10 hops), which is a commendable technical achievement.

3. Interpretability: The authors' effort to map the model's learned routing patterns back to established neurobiology—such as highlighting DMN and FPN dysfunctions in ASD patients—is highly appreciated and adds significant value to the paper.

**Weaknesses:**
1. Baseline Fairness (Text Modality): The model leverages a pre-trained language model to encode rich textual descriptions into prompts. It is not entirely clear if the standard graph baselines (like LSTMGCN or EV-GCN) were given access to these exact same high-quality textual embeddings. If they were not, the performance gap might simply reflect the representational power of the language model rather than the superiority of the proposed graph routing.

2. Overfitting Risks: Like many Transformer-based multimodal architectures, PhenoBrain is a very heavy model. Applying such a complex architecture (Language Models + Graph Transformers + FiLM modulation) to relatively small cohorts like ABIDE and PIXAR carries an inherent risk of overfitting that standard cross-validation may not fully capture without an independent external validation set.

3. Missed Opportunity in the Impact Statement: The paper provides a boilerplate statement claiming no potential societal consequences. Given the context of clinical AI, autism diagnosis, and the well-documented demographic biases in neuroimaging datasets, this feels like a significant oversight.

---

> ### Author Rebuttal · Authors · 2026-03-31
>
> 1.About Baseline fairness:
>
> We appreciate the reviewer's insightful observation. In fact, to ensure fairness, we provided identical text modality support for all benchmark models involved in the comparisons across all experiments. Specifically, we used the same frozen pre-trained language model (BERT) as PhenoBrain to extract globally pooled text vectors, and concatenated them with structured tabular features. For swarm graph models such as EV-GCN, this vector was used to calculate pairwise similarities between subjects, thus defining the edge weights of the swarm graph. For multimodal benchmarks such as MGDR, the text embeddings were entered as independent modal inputs into their respective dissociative representation learning or latent graph learning modules. We promise to complete these experimental details in the final draft.
>
> 2.About Overfitting Risks
>
> In fact, PhenoBrain essence is not to win by stacking parameters, but by using mechanisms to constrain overfitting, specifically in the following 2 aspects.
>
> (1) Our language models in PhenoBrain exist in a frozen form, meaning that this part only serves as a feature extractor, which effectively avoiding the overfitting risk common in large-scale language models.
>
> (2) PhenoBrain's core component, i.e., PCLR, employs Low-rank adjustment technology. In equ 5,  the rank $r$ is set to $16$, much smaller than the number of ROIs(200). This design mathematically forces an extremely narrow communication channel, compelling the model to learn only the most robust, restricted communication patterns.
>
> In addition to the above mechanistic explanation, we further supplement this with cross-site validation experiments. Specifically, due to the scarcity of neuroimaging data limiting its scale, we chose to conduct LOSO validation experiments on the ABIDE, comparing our method with the SOTA baseline method, MGDR. Due to space limitations, we only report the ACC and AUC metrics here. Experimental results show that in LOSO mode, PhenoBrain's ACC and AUC are 0.7347 and 0.7832, respectively, only $3.77%\$ and $2.3\%$ lower than the mixed site model. MGDR's ACC and AUC in LOSO mode are 0.6954 and 0.7749, respectively, lower than our model. This strongly demonstrates PhenoBrain's generalization robustness.
>
> 3.About Impact Statement
> Thank the reviewers for pointing out it! We have revised this section. Specifically, we will explicitly point out the feature imbalance in the dataset (e.g., the male proportion in ABIDE is approximately $85.2%$). This bias may cause AI models to favor different phenotypes, thereby increasing the risk of missed or misdiagnosed cases in minority groups. Then, we will emphasize that clinical AI should not be a "black box." Through the proposed long-range routing and attention modulation mechanism, the model provides a neurobiologically verifiable explanation.
>
> 4.About Multi-hop Performance
> PhenoBrain's PCLR mechanism has a computational complexity of $O(K|E|d)$ at hop counts of $K=10$. Since the brain network graph is sparsified based on a threshold $\tau$, its computational cost is significantly lower than the standard Transformer $O(N^2 d)$. On an  RTX 5090, PhenoBrain's measured memory usage during inference is 4.1 GB, while the standard 3-layer GCN is 1.8 GB; the single-subject inference time is 12 ms, and the 3-layer GCN is 9 ms. Although $K=10$ increases the computational cost by a constant factor, it gains the ability to model long-range collaborations in brain regions, which is diffucult to the standard GCNs.
>
> For the hop count in Figure 3, we supplemented the variance metrics to verify the stability of the performance improvement. Due to space limitations, we report only partial data. On the PIXAR dataset, the AUC with $K=9$ is $0.9259/pm = 0.0644$, a significant improvement compared to $0.9096/pm = 0.0563$ with $K=1$. On the ABIDE dataset, the AUC with $K=9$ is $0.8346/pm = 0.0498$, a significant improvement compared to $0.8031/pm = 0.0336$ with $K=1$. All experimental results will be reported in the final version.
>
> 5. About Limitations
> As the reviewers pointed out, we will add a rigorous disclaimer regarding biomarkers in the paper.  While the attention hotpots generated by PhenoBrain have been validated in existing studies, they are correlation inferences, not causal biomarkers. We explicitly remind readers that these findings should be considered clues for future interventional research, not the absolute gold standard for clinical diagnosis.
>
> Limitations: Although our method achieves a linear time complexity of $O(K|E|d)$, its memory consumption and computational latency are indeed higher than standard 3-hop GCNs due to the 10-hop intermediate representation storage involved. We acknowledge that the cascading of cross-hop features leads to a linear increase in memory usage with the value of K. We will suggest in the paper that future research can further optimize this bottleneck through sparse attention sampling mechanisms.

---

> > ### Author Rebuttal · Reviewer_RASD · 2026-03-31
> >
> > Thank you for the excellent rebuttal. This has fully resolve my concerns, so I'm raising the score.

---

### Official Review · Reviewer_hQyP · 2026-03-12

**Soundness:** 4
**Presentation:** 3
**Significance:** 3
**Originality:** 4
**Overall Recommendation:** 5
**Confidence:** 5

**Summary:**

This paper proposes a phenotype-based multimodal brain network analysis framework called PhenoBrain. This framework utilizes structured phenotypic and multimodal data as context to represent individual differences, thereby guiding the model to transmit information and select evidence within the brain network. Experimental results demonstrate that the proposed method achieves good performance.

**Compliance With Llm Reviewing Policy:**

Affirmed.

**Final Justification:**

My concerns have been fully resolved. The author's responses strengthened my acceptance of the paper. Overall, PhenoBrain is a technically solid work in multi-modal brain network analysis.

Therefore, I maintain my recommendation.

**Key Questions For Authors:**

In Equation 23, was the hyperparameter $\lambda$ determined through grid search? If the authors could share the model stability of this parameter under different values, it would be instructive for future researchers.

**Limitations:**

yes

**Strengths And Weaknesses:**

Strengths:

1.This paper proposes a novel approach that shifts from "decision fusion" to "mechanism fusion." By introducing phenotypic information during the representation learning stage rather than the classifier stage, PhenoBrain provides a highly inspiring architectural reference for multimodal brain network analysis.

2.The manuscript's diagram design is excellent. The motivational illustration in Figure 1 and the overall framework diagram in Figure 2 are very intuitive, helping readers quickly understand how multimodal information interacts and modulates within the model.

Weakness:

1.Equation 22 introduces the InfoNCE alignment loss, which forces brain representations to align with phenotypic representations. However, in clinical practice, there are often inconsistent cases such as "phenotypic abnormalities but delayed imaging manifestations" or "compensatory brain connectivity." The author need to discuss that.

2.In typical brain network analyses, the number of ROIs, N, is usually 116 (AAL) or 200 (CC200). Since each subject $s$ has its unique modulation term, this means the model needs to dynamically generate or map high-dimensional interactions for each individual. The authors need to discuss the range of choices for the rank $r$.

3.In Section 4.2.1, the authors provided a very detailed description of the experimental results. Transforming some of the repetitive textual descriptions into a more concise summary would make the paper's argumentation more robust.

---

> ### Author Rebuttal · Authors · 2026-03-31
>
> 1.Regarding the situation of "phenotypic abnormalities but delayed imaging manifestations" or "compensatory brain connectivity"
>
> We appreciate the reviewer's insightful observation regarding clinical-radiological dissociation and compensatory connectivity. PhenoBrain is specifically designed to accommodate these nuances through two core mechanisms, including (1) Probabilistic Manifold Alignment (Eq. 22): Unlike rigid identity-based mapping, our InfoNCE alignment encourages the model to seek a latent manifold where discriminative neural signals are coupled with clinical narratives. This probabilistic framework effectively handles "delayed imaging manifestations" by filtering out disease-irrelevant variance and focusing on robust neural signatures; (2) Subject-Specific Communication (PCLR): To address "compensatory brain connectivity," the PCLR mechanism generates individualized routing kernels modulated by $z_{pheno}^{(s)}$. This allows PhenoBrain to explicitly differentiate between connectivity signaling primary deficits (e.g., weakened DMN communication) and connectivity representing successful adaptation strategies (e.g., RIFG hyperconnectivity in ASD ), rather than enforcing a uniform population-level kernel.
>
> 2.About the range of rank r：
>
> In fact, in our method, the rank $r$ is not used as a hyperparameter, but rather as an approximation obtained through sparse-precision matrix estimation and singular value decomposition (SVD) of the brain network. Empirical studies of low-rank matrix factorization in neuroimaging show that the effective rank of the functional connectivity matrix is ​​typically much lower than the number of ROIs. In the PhenoBrain implementation, for the CC200 atlas with $N=200$, we determined $r=16$ to be the optimal choice through sparse-precision estimation and SVD of the functional connectivity matrix.
>
> Furthermore, we performed a parameter sensitivity analysis on $r$ on the ABIDE dataset. Specifically, we made $r \in [4,8,16,32,64]$ and reported the ACC metric. Due to character limitations, we will directly report the results. The results show that our method has ACC values ​​of 0.7212, 0.7347, 0.7635, 0.7614, and 0.7488 for the five values ​​of $r$. Experimental results indicate that if $r$ is chosen too low (e.g., $r = 4$), the model lacks the ability to capture subtle differences in inter-network interactions, such as subtle variations in the frontoparietal control network (CON) relative to the default mode network (DMN). Conversely, if $r$ is chosen too high (e.g., $r = 64$), performance also degrades. This is because the learnable parameters in the projected weights $W \in \mathbb{R}^{r \times d_c}$ increase significantly, leading to a high risk of overfitting on small cohorts common in neuropsychiatric research.
>
> 3.Regarding the improvements to Section 4.2.1.
>
> We sincerely appreciate the reviewer’s constructive feedback on streamlining our results presentation.  The revised Section 4.2.1 is provided below:
>
> "As summarized in Table 1, PhenoBrain consistently establishes a new state-of-the-art across all metrics. On the ABIDE dataset, PhenoBrain outperforms the strongest baseline (MGDR) by 3.46% in AUC, with significantly improved specificity. This suggests that by capturing idiosyncratic neural signatures through mechanism-level modulation rather than simple feature concatenation, our framework achieves more balanced decision boundaries in psychiatric diagnosis. For the PIXAR dataset, our model yields a 3.62% sensitivity gain over the best baseline (SPromptGL). This indicates that phenotype-conditioned routing is uniquely effective at identifying the subtle maturational shifts in functional connectivity that characterize brain development. Ultimately, the robust performance across both ASD diagnosis and developmental staging confirms that PhenoBrain provides a superior paradigm for evidence-based neuroimaging analysis."
>
> 4.About the analysis of parameter $\lambda$
>
> To investigate the model's sensitivity to parameters, we performed a comprehensive grid search on the ABIDE dataset for $\lambda \in \{0.001, 0.01, 0.1, 1.0, 10.0\}$. Due to character limitations, we can only report the ACC results; the remaining results will be included in the appendix after acceptance. The results show that our method's ACC for the five $\lambda$ values ​​are 0.7524, 0.7586, 0.7635, 0.7608, and 0.7537, respectively. The results indicate that our method is robust to the overall choice of $\lambda$, exhibiting high stability when $\lambda$ is around 0.1. When $\lambda$ is too small, the alignment signal is insufficient to guide the brain encoder, leading to a lack of clinical context and a decrease in classification accuracy of approximately 1.5%. When $\lambda$ is too large, the model overemphasizes alignment at the expense of class discriminative power, resulting in an increase in the variance of the final prediction.

---

> > ### Author Rebuttal · Reviewer_hQyP · 2026-04-02
> >
> > My concerns have been addressed. I will keep my positive score.

---

### Official Review · Reviewer_wtSn · 2026-03-12

**Soundness:** 3
**Presentation:** 3
**Significance:** 4
**Originality:** 3
**Overall Recommendation:** 5
**Confidence:** 4

**Summary:**

The paper presents and evaluates a new algorithm for multi-modal brain network analysis. The method nicely uses phenotypic information as conditional priors to modulate brain networks. Accuracy is demonstrated to be mostly improved relative to other methods.

**Compliance With Llm Reviewing Policy:**

Affirmed.

**Final Justification:**

The paper presents and evaluates a new algorithm for multi-modal brain network analysis. Responses from the authors also address my concerns.

**Key Questions For Authors:**

1.The processing of clinical texts appears to pose a potential risk of data leakage. Have the authors addressed this issue and developed a solution?
2.The author currently primarily uses the CC200 template. Does this framework also support other commonly used anatomical templates (such as the AAL template)? Can the long-range routing pattern remain consistent across templates of different levels of detail?

**Limitations:**

See the weaknesses

**Strengths And Weaknesses:**

Strengths:
1.This paper is well written and organized.
2.The proposed PhenoBrain are technical solid, which injects phenotype information at the mechanism level rather than only at the classifier level.
3.Extensive experiments show the effectiveness and robustness of the propsed method.

Weakness:
1.The processing of clinical texts may pose a potential risk of data leakage, and the authors need to discuss this.
2.The figures and tables in the paper are very valuable, but the font size of the axis labels and legends in Figures 3 and 6 is small, which may make them difficult to read in the printed version. It is recommended to increase the font size of these elements to improve the reading experience.
3.Figure 6 shows the top 20 key ROIs. Adding a short paragraph in the text explaining the anatomical names corresponding to these ROI indices (such as the prefrontal cortex, parietal cortex, etc.) would make the biological significance of the conclusions more intuitive.

---

> ### Author Rebuttal · Authors · 2026-03-30
>
> 1. About Potential Risk of Data Leakage in Clinical Text Processing
>
> Response: Thanks for raising this important question! We have implemented sufficient safeguards in our paper to ensure that the use of text data will not lead to data leakage. Specifically, during the text processing stage, we paid particular attention to removing all terms directly related to clinical diagnoses (such as "ASD," "diagnosis," and total scale score). These terms often become obvious predictive signals for the model, so we strictly masked these keywords during data preprocessing. Furthermore, we adopted standard text preprocessing methods, including removing irrelevant noise and normalizing text content. Through these processes, we ensured that the model relies only on behavioral and linguistic features in the text description, thus avoiding potential data leakage risks.
>
> 2. About Font Size in Figures and Tables
>
> Response: Thanks for your attention to the figures and tables! We will modify the font size in Figures 3 and 6 based on your suggestions to ensure that the axis labels and legends are clearly legible and maintain good visibility on different sized devices and printed versions. The revised version will be updated in the final submission of the paper.
>
> 3. About the Anatomical Names of the First 20 Key ROIs in Figure 6
>
> Response: Thanks for the reviewer's suggestion! We will add a short text to the paper explaining the brain region names corresponding to each key ROI listed in Figure 6. This includes the medial prefrontal cortex, posterior cingulate cortex, and bilateral temporoparietal junction in the precuneus of the DMN network; the dorsal anterior cingulate gyrus, insula, and prefrontal cortex in the CON network; and the inferior frontal gyrus, middle temporal gyrus, superior temporal sulcus, and parahippocampal cortex in the FPN network. The revised version will be updated in the final submission of the paper.
>
> 4. About the Anatomical Brain Atlas Template
>
> Response: Thans for the reviewer's question! Currently, our framework mainly uses the CC200 template (200 brain regions), but it is actually highly flexible and supports other common templates, including AAL. Specifically, we conducted a 10-fold cross-validation experiment on the ABIDE dataset using the AAL Atlas template (116 brain regions) and Craddock 400 CC Atlas template (400CC). At result, the AAL reach ACC of 0.7452±0.0417 and an AUC of 0.8137±0.0258, the Craddock 400 CC reach ACC of 0.7724±0.0282 and an AUC of 0.8407±0.0373. The results show that the high-resolution template achieved better performance than the low-resolution template without causing significant performance fluctuations. Furthermore, the main idea of ​​the long-range routing pattern is to dynamically adjust the communication strength between brain regions based on individual phenotypic information. Therefore, we observed that despite differences in specific representations between templates, PhenoBrain's long-range routing pattern effectively captures individualized neural network features under each template.

---

> > ### Author Rebuttal · Reviewer_wtSn · 2026-04-01
> >
> > The responses addressed my concern, and I am willing to remain my positive score.

---

### Decision · Program_Chairs · 2026-04-30

**Decision:**

Accept (spotlight)

**Comment:**

This paper proposes a framework for multi-modal brain network analysis by injecting phenotype information at the mechanism level. Its contributions include a phenotype-conditioned long-range routing mechanism and a phenotypic-guided attention mechanism regulation method. Experiments justify its superiority. All four reviewers provide positive feedback by recognizing the technical contributions and extensive experiments. Reviewers point out that the significance and originality are remarkable. The reviewers’ concerns are fully resolved in the rebuttal.